# Solving Multi-agent Path Finding as an LLM Benchmark: How, How Good and Why

**Weizhe Chen** *weizhech@usc.edu*
*University of Southern California*

**Sven Koenig** *svenk@uci.edu*
*University of California, Irvine*

**Bistra Dilkina** *dilkina@usc.edu*
*University of Southern California*

**Reviewed on OpenReview:** *https://openreview.net/forum?id=8hAxEFRVQT*

## Abstract

The rapid success of large language models (LLMs) has spurred extensive research into their ability to solve a wide range of tasks. However, their potential in multi-agent planning remains underexplored. Multi-agent planning presents unique challenges due to the combined complexity of coordination and long-horizon reasoning, often making it difficult to leverage external tools for assistance. In this paper, we introduce Multi-Agent Path Finding (MAPF), also known as multi-robot route planning, as a novel benchmark for evaluating the reasoning capabilities of LLMs. We first describe how the MAPF benchmark can be adapted for LLM-based evaluation, including dataset curation and an agentic workflow for LLMs. We show the motivating success of single-agent planning and multi-agent pathfinding in an empty room map without obstacles, then the failure to plan on the harder room map and maze map of the standard MAPF benchmark. We present our position on why directly solving MAPF with LLMs has not been successful yet, and we use various experiments to support our hypothesis. Based on our results, we discussed how researchers with different backgrounds could help with this problem from different perspectives.

## 1 Introduction

Since ChatGPT came out, large language models (LLMs) have been shown to go beyond strictly language-related tasks like translation, and to be a powerful tool in all kinds of domains. Training with very rich and diverse datasets, LLMs incorporate a large variety of knowledge and do not require fine-tuning before generating good solutions in many real-world applications. Over time, recent studies have shown that combining LLMs with different ways of prompting can help solve problems that have some aspects of reasoning, including examples from logical problems Yao et al. (2022); Liu et al. (2023) to controlling a robot dog without finetuning Wang et al. (2023a).

Motivated by the success of LLMs, people are studying how well pre-trained models perform in all different kinds of domains, and propose benchmarks. Some recent works studied the performance of LLMs on multi-agent problems and showed LLMs can also help multi-agent coordination Chen et al. (2023b); Agashe et al. (2023). However, they barely cover multi-agent route planning and do not look specifically into the difficulties in the domain. In this paper, we consider the problem of multi-agent path finding (MAPF), also known as multi-agent route planning. MAPF is the problem of moving a group of agents from their respective start locations to their goal locations without collisions, and can be used directly to formulate real-world

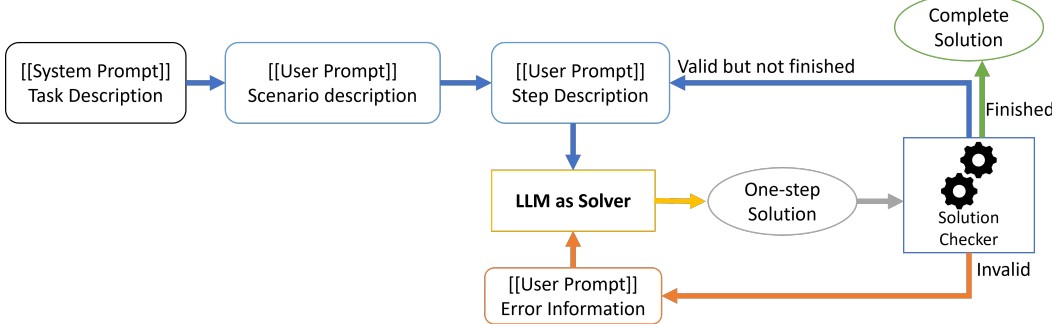

Figure 1: An illustration of our workflow.

applications like warehouse management Sharon et al. (2015); Han & Yu (2020), swarm control Li et al. (2020), among others. In a typical warehouse scenario, around a thousand warehouse robots (agents) could be running simultaneously in one warehouse room (scenario), and each agent needs to plan its path, which could be as long as a hundred timesteps.

MAPF is a unique multi-agent coordination problem where coordination must be embedded within the planning process. This makes it particularly challenging to enable LLMs to reason about coordination effectively. While using LLMs to generate high-level guidance combined with a low-level path planner has been effective in environments like Overcooked Agashe et al. (2023), this approach is unlikely to work for MAPF, as effective coordination must be tightly integrated into the planning itself rather than treated as a separate guidance layer. On the other hand, path planning is one of the easiest aspects of finding a solution in planning problems, as a special case compared to more general problems like Blocksworld Valmeekam et al. (2024a), due to its more interpretable state space and constraints that only involve obstacle avoidance and movement continuity. Due to the unique challenges inherent in the MAPF problem, the effectiveness of LLMs in solving MAPF remains uncertain.

In this paper, we aim to introduce the Multi-Agent Path Finding (MAPF) problem to the machine learning community as a new benchmark, with a particular focus on the reasoning-intensive setting of solving MAPF using large language models (LLMs) directly—without relying on traditional heuristic-based solvers or any additional task-specific training. We test the LLMs by giving the scenario information to the LLM, and let the LLM recommend the actions for each agent every single timestep. We use a high-level checker that checks collisions between agents and obstacles to ensure the solution is valid in each step, provides error messages to the LLM when violations occur, and iterates with the LLM until a valid solution is generated. We first show that single-agent pathfinding can be solvable and then evaluate the performance of LLMs on the empty map, the room map, and the maze map, which are classic maps from the standard MAPF benchmark Stern et al. (2019). We show that LLMs can solve MAPF problems when the scenario is relatively easy, and LLMs fail to generate any valid solution, no matter what the solution quality is, when the scenario becomes harder. While LLMs are evolving extremely fast, and current failures may not apply to the next generation of LLMs, we elaborate on the current failing scenarios and give our position on three aspects as a breakdown of what kind of capability is currently missing in the LLM workflow, namely 1) the capability to understand the scenario, 2) the context length limit, and 3) the reasoning capability. We incorporate a list of experiments featuring various prominent prompt design alternatives, such as image-based and text-only inputs, and with or without single-step observation information, among others. Finally, we provide a discussion on the challenges of using LLMs for MAPF in the real world, and discuss how different researchers could contribute to the problem from different directions.

We hope our work can serve as a building block for future research in foundation models for MAPF. And more importantly, we hope this paper introduces LLM researchers to the unique and persistent challenges of MAPF. We envision MAPF becoming a benchmark for evaluating future LLM advancements, particularly in methods addressing the failure modes identified in this study.

# 2 LLMs for Multi-agent Path Finding

## 2.1 Preliminary

The multi-agent path finding (MAPF) problem is the problem of finding a set of collision-free paths for a set of agents in a known environment while minimizing their travel times. Specifically, in this paper, we consider the problem in Stern et al. (2019); Li et al. (2022), which is a four-connected grid map, where each agent is given a start cell and a goal cell. A scenario is defined as the combination of the description of the map, indicating which cells have obstacles, and the start cells and goal cells of each agent. At each timestep, an agent can move to an adjacent cell or stay in its current cell. A collision happens if two agents end up in the same cell at the same timestep. Each agent remains at its goal cell after it arrives until all agents arrive at their goals. The objective is to minimize the makespan of the solution, i.e., minimize the time when the last agent arrives at its goal location.

With the number of parameters of LLMs being exponentially larger than the degree of freedom in small reasoning problems, LLMs have the potential to solve some easy problems in reasoning with certain prompts Wei et al. (2022), and break down harder problems into smaller ones to get suboptimal solutions Yang et al. (2023). When applying to MAPF, we hope LLMs can be an alternative model to the current MAPF reinforcement learning-based models without any additional training.

## 2.2 Benchmark Protocol

### 2.2.1 Dataset Files

Multi-Agent Path Finding (MAPF) is a well-established research domain within classical planning, supported by a comprehensive benchmark dataset introduced by Stern et al. Stern et al. (2019).

The dataset comprises 33 maps, each provided both as an image and as a corresponding symbolic textual representation, where '@' denotes fixed obstacles and '.' represents free cells. For each map, two class of scenario files are available, corresponding to different agent distribution strategies: (i) agents distributed uniformly across the map, and (ii) agents placed based on random coordinate generation, resulting in a less uniform spatial distribution. By default, we will use the first setting in this paper. Each scenario file contains a list of start and goal positions for agents. When evaluating a scenario with $n$ agents, the first $n$ entries in the scenario file are used to specify the corresponding start and goal locations. For repeated evaluations on the same map with different start-goal combinations, scenario files are used in sequential order, beginning with scenario file 1.

The maps in the dataset can be categorized into several distinct layout types, each presenting unique challenges and requiring different planning capabilities. Every map is assigned a descriptive name that encodes its structural characteristics. For instance, "Maze-32-32-2" indicates that the map belongs to the maze class, has dimensions of $32 \times 32$, and is the second instance of this type in the benchmark. In this work, due to the current limitations of LLMs in complex spatial reasoning tasks, we focus our experiments on three representative maps: EMPTY-8-8, ROOM-32-32-4, and MAZE-32-32-2. We encourage the community to also use the same map we are using as a starting point, and use the remaining maps and more scenarios when the capability of LLMs becomes better. For potential training, the map can be used, but the scenario files should not be directly trained on to avoid overfitting.

### 2.2.2 Evaluation Metrics

The metrics of the problem primarily consists of success rate and solution quality. Success rate is about whether the solver can find a complete solution for the given scenario that is collision-free and every agent has reached their goal. The solution quality is defined as the makespan, i.e., the time spent from when all the agents start moving to when the last agent reaches its goal. There is also the variant of the sum of cost, defined as the total sum of time all agents spent to reach their goal, respectively. Given the current success rate of LLMs, we will leave the solution quality metrics for future research.

While the objective of the problem is to let LLM solve the problem through the workflow, we cannot allocate infinity budget on the problem and infinitely loop in the magnetic workflow. Thus, we define a solution generation to be successful if it does not fail because of any of the following, and early-stop the generation correspondingly:

1. Fails to generate a plan whose number of steps is at most 3 times that of the optimal plan.

2. Fails 5 consecutive times in a single step after we provided the first round of feedback, specifying whether the current solution is correct or not.

### 2.3 Methods

In this paper, we focus on using LLMs to solve the MAPF problem directly. However, it is obvious that current ML models cannot be perfect solvers in their first trial, and we introduce a high-level collision checker to ensure the plan generated by the LLM is valid. We inform the LLM about the mistake in the current solution, if any. Unlike some previous works Yao et al. (2022); Yang et al. (2023), our checker is not another LLM both because it is extremely easy and efficient to detect collisions by a rule-based detector in linear time, and also because the LLM fails to always correctly identify the collisions. A detailed comparison is provided in the appendix. In this checker, we not only check for agent-to-agent collisions but also check for any collision with the fixed obstacles. We do not provide any additional guidance on how the LLM should resolve this. By default, all the information, like coordinates, is provided in text, and we will later discuss the performance difference caused by different input formats.

Following the common practice of LLMs Kambhampati et al.; Chen et al. (2023b), we build an agentic workflow shown in Fig. 1. As existing learning-based approaches Damani et al. (2021), we give LLMs stepwise local information and let the LLM choose the actions of agents step-by-step. This step-by-step (SBS) generation is different from the popular chain-of-thoughts idea Wei et al. (2022) used in LLMs by not introducing more intermediate reasoning processes in the generation. Instead, it breaks down the whole planning task into smaller single-step tasks, so the LLM does not need to be fully correct before we can use some results from it. Later in the paper, we compare SBS with a variant that generates the whole plan in one shot (OS) with the help of chain-of-thought prompts and show the comparison in Table. 2. We start by giving the LLM the system prompt to become a solver for the MAPF problem. This part is fixed within each map, and differs in the map description part in different maps. Then, we start the user prompt by providing the scenario information, which includes where the obstacles are, as well as the start location and the goal locations of each agent. We found that providing LLMs with specific local information about viable actions for the immediate next step significantly helps them avoid collisions with static obstacles, so we provide a single-step observation (SSO) in the prompt by default. Then we start to read the output of the LLM, use the checker to determine whether the output of the LLM is valid, and either output the errors to let the LLM correct on the current step, or tell the LLM to move on to the next step with the information that is specifically for the next step. To address the total token limit, we leverage the fact that MAPF is a Markov decision process where each state is independent of its previous states, and we restart the prompt from scratch, i.e., treat the current agent locations as their starting point, whenever the current context to be fed into the LLM is longer than the model limits.

Although there are numerous possible strategies for constructing prompts for LLMs, we defer a detailed discussion to the next section, where we use different prompt designs to help analyze the causes of failure. The prompts used in our experiments are provided in the appendix.

### 2.4 Benchmarking Results

### 2.4.1 General Experiment Settings

Each setting in the experiment is tested on 5 different scenarios in the standard MAPF benchmark Stern et al. (2019). By default, we are using the GPT-4-1106-preview model, also known as the GPT-4-turbo model with temperature 0 and seed 42.

| Map | Dist | Success Rate (%) | | |
|-----|------|------|-----|-----|
| | | OS | SBS | A* |
| Room | [2, 8) | 80 | 100 | 100 |
| | [8, 16) | 40 | 100 | 100 |
| | [16, 32) | 40 | 80 | 100 |
| | [32, 100) | 20 | 40 | 100 |
| Maze | [2, 8) | 20 | 80 | 100 |
| | [8, 16) | 0 | 20 | 100 |

Table 1: The success rate of solving single-agent path finding with GPT-4-turbo on ROOM-32-32-4 and MAZE maps with varying ground-truth shortest path distance (Dist). OS and SBS are different prompt versions for LLMs, while A* is a search-based baseline that is purely included to demonstrate the failure of LLMs compared to previous algorithms. The names of the methods are simplified. For OS, it is combined with GO and TOM, and for SBS, it is combined with GO+SSO and TOM. All the details for these settings are described later.

| Map | n | Success Rate (%) | | |
|-----|---|------|-----|-----|
| | | OS | SBS | CBS |
| Empty | 2 | 20 | 100 | 100 |
| | 4 | 0 | 100 | 100 |
| | 8 | 0 | 100 | 100 |
| | 16 | 0 | 60 | 100 |
| Room | 2 | 20 | 100 | 100 |
| | 4 | 0 | 80 | 100 |
| | 8 | 0 | 20 | 100 |
| Maze | 2 | 0 | 0 | 100 |

Table 2: The success rate of solving MAPF with GPT-4-turbo on EMPTY-8-8, ROOM-32-32-4 and MAZE-32-32-2 maps with varying numbers of agents (n). OS and SBS are different prompt versions for LLMs, while CBS is a search-based algorithm that is commonly used in MAPF. For OS, it is combined with GO and TOM, and for SBS, it is combined with GO+SSO and TOM. All the details for these settings are described later.

We do not consider any token limit error in this paper because we restart at each step to clean its travel history if the token limit is too long, and 5 times in a single step guarantees that the message length in one step is within the token limit.

### 2.4.2 Single-agent Path Finding Results

As discussed in previous work Valmeekam et al. (2024a), while LLM might not know how to generate an optimal plan, our results show that it could generate some plan regardless of the optimality. As shown in Table. 1, we found that LLM could generate a plan after a few iterations in relatively easy scenarios. We observe that with the growing groud-truth shortest path length, the success rate of LLM drops significantly, especially in the one-shot settings (OS). In the maze scenario, it is hard to find one solution even in the smallest case, and we will discuss the reasons in the next section. In contrast, traditional search-based algorithms like A* solve the problem at the same scale very easily.

We also found that the cause of failing in pathfinding is different from the cause of failing in general planning tasks where the generated plan is invalid, usually by breaking requisite constraints Valmeekam et al. (2023). In path planning, the only constraint is that one agent needs to move to an adjacent cell and cannot jump across the map. This is a very easy task for LLM to follow, and thus, in our experiments, we never see the LLM failing due to that constraint, even if the only instruction about this constraint is "in a 4-connected grid". Building on our findings in single-agent path planning, we posit that large language models (LLMs) hold promise for generating effective solutions to multi-agent path finding (MAPF) problems. In particular, small-scale MAPF scenarios may serve as both a practical application domain and a distinctive benchmark for evaluating the reasoning and planning capabilities of LLMs.

### 2.4.3 Multi-agent Path Finding Results

We start by showing that LLMs can be a promising solver in easy domains. We test the performance of LLMs on the EMPTY map from the MAPF benchmark Stern et al. (2019), which has no obstacles at all and only requires the agents to not collide with each other. The results are shown in Table. 2. We observe that for smaller-scale problems that have fewer agents, GPT-4 can successfully generate valid solutions with the workflow. However, GPT-4 with the same workflow fails to generate valid solutions when more agents are on the same map, while classic algorithms like conflict-based search (CBS) easily maintain a perfect success rate in solving the problem. We also observe that one-shot (OS) generation fails to generate solutions even

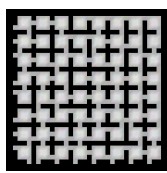 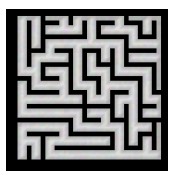

Figure 2: The ROOM-32-32-4 map (left) and the MAZE-32-32-2 map (right). The picture is vertically flipped to match the common knowledge that higher vertical positions indicate greater values.

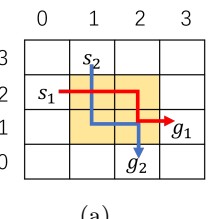 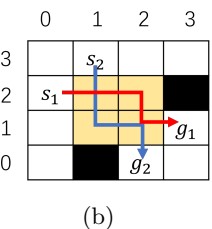

(a)                              (b)

Figure 3: Two examples of the MAPF problem, which are also known as the *symmetry breaking* examples for MAPF community, where black denotes a fixed obstacle. In 3a, every pair of shortest path will collide with each other. In 3b, every pair of shortest paths will collide in cell (2, 1).

in scenarios with 4 agents. This is because the whole plan is quite long, and the total number of possible failures is the product of possible failure at each step, making it inefficient to find a valid solution.

Motivated by the success on the EMPTY map, we tested the same workflow on the ROOM-32-32-1 map and the MAZE-32-32-2 map, which is shown in Fig. 2. The results are shown in Table. 2. In this case, we found that LLMs fail to generate solutions when the number of agents is already quite small, and that LLMs completely fail on the MAZE-32-32-map even with only 2 agents. Motivated by the superior performance of SBS over OS, we adopt SBS for the remainder of the paper without further specification.

In our experiments, we observe that LLMs fail in the empty map scenario with 16 agents due to a rare case of consecutive missteps in a single planning step. In all other cases, failures arise from overly long solution paths. Specifically, 77% of the failures are caused by oscillatory behavior, where agents repeatedly move within a confined area, while the remaining failures result from unnecessarily long detours.

## 3 Cause of Failures

Given the rapid pace of advancement in large language models, it remains uncertain how long current limitations will persist. Nonetheless, systematically analyzing these failures and identifying their root causes is essential for progress. Specifically, we categorize the failure modes into three underlying sources from the perspective of natural language models: general reasoning capability, long-context capability, and understanding symbolic information.

### 3.1 Reasoning Capability

In this paper, we focus on optimizing the makespan of the MAPF solution, and making the agents prefer to wait rather than detour in small scenarios that do not lead to endless waiting. We calculate the average ratio of makespan to the maximum length among the single agent shortest paths (a standard normalization practice in MAPF as this provides a valid lower bound on the optimal MAPF solution), and in scenarios that are successfully solved as denoted by the ones in Table 2, this average is 1.5. This means that LLMs can mostly succeed in scenarios that do not need a lot of waiting, and in most steps, they only need to go in the two directions aligned with the goal's direction. LLM fails in other scenarios because it needs pathfinding in complex environments. A simple example of the failure of path finding is shown in Fig. 3, and we provide the output of the first step in Fig. 4. While waiting for one step can clearly lead to the optimal solution that has a makespan of 5, the LLM chose to move agent 2 to the left and resulting in a total makespan of 6. This example shows that LLM does not understand what makespan is and how to optimize for makespan.

On the other hand, when we look at the average number of iterations for prompts with both global obstacle observation in the first user prompt and single-step obstacle observations(GO+SSO) in Table. 3, it is still not very close to 1 which means no iterations at all. This is because even if we have explicitly let the LLM list all the coordinates of agents, it does not have the capability to check the answers itself without the use

```
Step 1:
Agent 1:
Current: (0,2)
Goal: (3,1)
Move: Right to (1,2)

Agent 2:
Current: (1,3)
Goal: (2,0)
Move: Down to (1,2)

After the move, the coordinates of the agents are:
Agent 1: (1,2)
Agent 2: (1,2)

There is a collision between Agent 1 and 2. I will regenerate another solution immediately.
...
Agent 2:
Current: (1,3)
Goal: (2,0)
Move: Left to (0,3)
...
```

Figure 4: Part of the output from GPT-4-turbo for the symmetry breaking example in Fig. 3b. Full example is provided in the appendix.

of external tools. More specifically, LLMs are not guaranteed to tell whether a list of tuples of numbers has any duplicates, and can often fail because of this. We also observe that the GPT-4-8K model requires more iterations compared to the GPT-4-Turbo model, which we attribute either to increased information retention challenges in the former or to the enhanced capacity of the latter.

In many problems, limited reasoning capabilities are often addressed through tool use. However, finding a good way to use tools in MAPF remains particularly challenging, as it requires not only pathfinding in a complex environment but also the strategic and efficient avoidance of collisions among all agent pairs. The coordination between agents is required in each step, and each step only. If we let the tool, in this case, a single-agent planner like A*, include collision avoidance, the problem is already solved as CBS Sharon et al. (2015), where LLM did no help. On the other hand, the tools must know how collisions are supposed to be avoided and add those constraints into the heuristic search algorithm. This paradox makes it very difficult to use LLM with tools in MAPF.

We have also tested the o1-preview model from OpenAI, which has been shown to significantly improve the reasoning and planning capability Valmeekam et al. (2024b). Compared to the standard rule-based checker, which is the default for all other experiments, we introduce a human checker that analyzes the explanation and chooses to retry when the current explanations could lead to future failure. We observe that such an improved checker can also facilitate the generation a lot, which further indicates that the current reward model used in training and, potentially, the inference is not correct for MAPF. Examples and more explanations on why o1-preview leads to a worse performance compared to GPT-4-turbo in the default settings are provided in the appendix.

## 3.2 Long-context capability

Because the underlying architecture of popular LLMs is transformers, which further rely on self-attention, a longer context in the input will significantly increase the computing complexity in the process. Therefore, in their training process, they set a maximum limit of tokens on the input and trained specifically on them. Current large language models released will also provide a context length limit, which is as long as 200K

| Model | n | Success Rate (%) | | Avg. Iterations | |
|---|---|---|---|---|---|
| | | GO | GO+SSO | GO | GO+SSO |
| GPT-4-8K | 2 | 80 | 100 | 2.7 | 1.6 |
| | 4 | 20 | 60 | 3.0 | 2.3 |
| | 8 | 0 | 0 | N/A | N/A |
| GPT-4-128K | 2 | 100 | 100 | 2.1 | 1.2 |
| | 4 | 60 | 80 | 2.7 | 1.4 |
| | 8 | 0 | 20 | N/A | 2.4 |

Table 3: The success rates and average iterations per step used until proceed to next step in success scenarios for GPT-4 and GPT-4 Turbo, whose token limits are 8K and 128K respectively, on the room-32-32-4 map with different number of agents(n). The name of the variants are simplified. Both variants are tested under SBS and TOM, with details described in other parts of the paper.

| Checker Type | Success Rate (%) |
|---|---|
| Rule-based | 0 |
| Human | 40 |

Table 4: The success rate of o1-preview (OpenAI) with different checker type on 8 agent scenarios ($n = 8$) with GO+SSO+SBS+TOM in room-32-32-4 map.

tokens, and any request with a longer context length will be rejected. This is a long enough length for many tasks and even able to read a textbook, and many users are satisfied with the length. Recent studies have demonstrated that the performance of large models, such as GPT-4 Turbo with 128K context length, is not consistent when processing inputs of 8K versus 128K tokens [1]. What is even worse is that in the real-world applications of MAPF, the total number of agents running in the environment could be up to a few thousand. Even in the ideal case that our prompt will grow linearly to the number of agents, an environment with a thousand agents will require a total context length of $250(tokens/agent\ step)*50(steps)*1000(agents) = 5M$ tokens, which is much bigger than the current limit even without considering the related reasoning capability under those length. Because the token limit is strict, we have introduced the restart mechanism in our methods, and in our current experiment scale, we have to restart the LLM around every 7-10 steps in large or complex scenarios. We also observed that the number of tokens used grows non-linearly with regard to the number of agents due to the increasing number of iterations, and we have put the numbers in the appendix.

In table. 3, we show our results on the ROOM map, and test with different GPT-4 models with different context lengths, in the setting of global observation (GO) only and the version with single-step observation (SSO). We will talk about these two settings in the next section. We found that when the context length limit grows, the success rate also increases. This is especially helpful when the single-step observation information that tells what valid actions each agent has is not provided. The failures here are not direct failures by exceeding the context length limit, but from the forgetting in restart caused by reaching the context length limits. Because of these restarts, our models often completely forget that they have been to certain locations, and then go back and forth in certain areas with dead ends like the center room in the ROOM map. We have also included the o1-preview model from OpenAI Jaech et al. (2024) in the experiment on the biggest scale. To our surprise, it is even worse than the GPT-4-turbo version. In the appendix, we provide a case study with more detailed explanations. And because the o1 models are equipped with the capability of writing programs that help them determine in the process, their failure of them also indicates that LLMs currently cannot naturally link the problem of MAPF as one of the problems they can write a program to solve, e.g., write a CBS algorithm and solve the problem. While manually instructing the LLMs to write a code of CBS is very easy for LLM [2], in this paper we focus on the underlying capability of LLMs to find a way to solve MAPF themselves without any guidance.

## 3.3 Understanding Symbolic Information

MAPF scenarios can be broken down into a pair of start location and goal locations, together with the map information. While understanding the coordinate version of start goal location pairs is relatively easy, understanding the map information is hard. Trained with publicly available text, LLM learns what a specific

---

[1] https://github.com/gkamradt/LLMTest_NeedleInAHaystack
[2] Latest LLMs have a 100% success rate when explicitly required to write the CBS algorithm.

| n | MM | TOO | TOM |
|---|----|-----|-----|
| 2 | 100 | 100 | 100 |
| 4 | 20 | 60 | 80 |
| 8 | 0 | 0 | 20 |

Table 5: The success rate (%) of different ways of inputting the map information to LLM on ROOM-32-32-4 map with different number of agents (n). All settings are combined with SBS+GO+SSO.

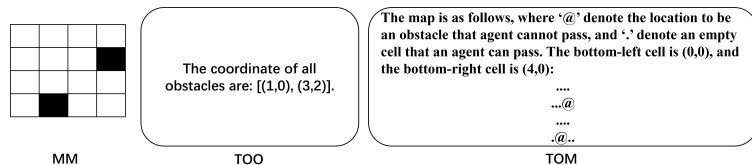

MM        TOO        TOM

Figure 5: An illustration of the difference between multimodal input (MM), text that describes the whole map (TOM), and text that describes the obstacles (TOO).

map means by finding similar contexts online, which mostly comes from other related fields in planning, like solving a maze. However, one problem is that people barely provide any such information online since people have the common knowledge of what to do with a map with code and preprocess the map information in the code rather than explicitly provide the set of where the obstacles are as the original input. Therefore, this leads to a lack of training data with related context about the information in the pictures or in a symbol-based input. As we mentioned earlier, 77% of the overall failure is caused by agents detouring back and forth in a certain area blocked by obstacles. These failures directly indicate that the LLM fails to understand the map information provided.

Because of this, our main prompt included guidance on what action could be taken in the current step to increase the success rate a little bit. In Table. 3, we find that while global observation can achieve a high win rate with the support of our high-level checkers, it also increases the average number of iterations per step. Furthermore, it fails in scenarios where SSO would be effective, as it exceeds the step limits defined in Sec. 2.2.2. In general, LLMs fail in medium complexity maps like ROOM, and completely fail in much harder maps like MAZE, even with the help of SSO.

In all the experiments above, we use pure-text-based input because of the popularity and simplicity of pure-text input. However, this was not the case before: In earlier work of reinforcement learning for MAPF, people transformed the input into image-based input within the field of view and used neural networks like convolutional neural network (CNN) to read the information. Therefore, as shown in Fig. 5, we propose three different methods, which all share the same prompt of describing the problem definition and use different versions of the prompt describing the scenario:

1. A multimodal-based method MM where we use an image to give the picture of the map, and text to describe the problem and the scenario.

2. A text-only method TOO that describes the map by listing the coordinates of the obstacles.

3. A text-only method that describes the map by feeding the text version of the map with '@' denoting the obstacles and '.' denoting an empty space, named TOM.

We test the performance of these three variants, and show the results in Table. 5. We found that with an image-based input, the results are even worse. For example, large models can only have a high-level idea that there is a wall in the middle area, rather than knowing there is a wall exactly from $(3, 10)$ to $(20, 10)$. The failure of TOO further indicates that LLMs do not know how to combine high-level intuitions of where obstacles are with concrete reasoning during planning. Overall, in complex environments such as mazes and real-world city maps, LLMs cannot offer any form of guidance, not necessarily the specific action recommendations tested in this paper, until they can accurately comprehend the relationship between obstacles and the paths of agents.

## 4    Discussion

We would like to note that the methods in this paper are not designed to either outperform the state-of-the-art classic solvers or surpass the performance of learning-based solvers. Instead, the goal is to demonstrate that LLMs can solve small problems simply through prompting and to analyze the factors limiting their ability to handle larger scenarios. Specifically, current heuristic-based algorithms like conflict based search Sharon et al. (2015) can solve all the scenarios tested in the paper in less than 0.1 second. Moving on, we believe the three reasons for failures can each independently relate to different research directions. Improving capability in a long context is already a popular direction in NLP, and many papers are also looking into improving the general reasoning capability. We believe that much ongoing research has the opportunity to improve the performance of LLMs on MAPF, and we are pleased to have more researchers test this unique problem. Also, the recent development of building foundation models that combine language and image generation could potentially help with the reasoning in MAPF. Image generation models like stable diffusion Rombach et al. (2022) have shown to be a strong tool in single agent path finding Janner et al. (2022). While these diffusion models are not designed for multi-agent planning and are even hard to apply to MAPF, strategically using them could partially help the lack of reasoning capability. Regarding potential finetuning, we believe more annotated data are needed to help foundation models connect the abstract description to concrete coordinate-level knowledge of where obstacles are, and how to coordinate between agents. Future research could finetune using their data, or at least open their data to the general public, and let the next version of released models from industry companies include such data by automatically getting data online. Hopefully, such data could help foundation models know that position encodings are more important in planning problems. Additionally, our results on current rule-based checker v.s. human checker also indicate that future research could work on building a good checker, i.e., a reward model in large language models, to improve the performance of LLMs. However, we still acknowledge the fact that while some certain more advanced way of solving the problems, e.g., more advanced prompts, cannot directly address all the obstacles mentioned in the paper, it is still possible that they can be beneficial to the domain and might lead to some additional gaps to be solved. Overall, we believe that as a unique domain, MAPF can be a useful benchmark for general LLM researchers to test their idea on.

While our current prompt has already included a lot of common tricks for prompting the LLMs, there are some other tricks that we found not as helpful as others on GPT-4-turbo, and we did not include them in the final version. Specifically, this include ReAct Yao et al. (2022), CoT Wei et al. (2022) upon SBS, vision of thoughts Wu et al. (2024), adding a whole example to the prompt as in-context learning examples, and breaking the instructions into bullet points.

Although the success rate is one of the most important factors in measuring the performance of a solver, there are also other challenges that need to be addressed before using LLMs in real-world scenarios. First of all, success does not indicate anything about the quality of the solution. In our experiment, we found that the success scenarios usually come from easier scenarios that agents mostly only need to follow their direct shortest path. When planning gets harder, the success rate gets much worse. To improve the reasoning capability, future researchers and users could choose to make a dataset of scenarios and their good solutions publicly available on the internet, so when training with newer data, this specific problem could be directly included in the training set. Furthermore, the current workflow does not include any heuristics or tool use during the generation but only as a solution checker. Although the best approach for incorporating these aspects remains uncertain, we believe this presents a highly challenging yet meaningful research direction. For potential researchers going in this direction, we want to remind them to keep looking at the consistency of the deconflict of agents, which could be seen as moving obstacles, and the deconflict of the fixed obstacles on the map. A successful method should consider both difficulties in a similar manner.

On the other hand, latency is always a problem for LLMs when used in real-world production. If one wants to use their own models, which is currently  500 tokens per second after using state-of-the-art system-wise optimization like vLLM Kwon et al. (2023) for LLAMA-2 Touvron et al. (2023). However, this speed is not enough as a typical round of output will have 1500 tokens, and it may take a few rounds before the LLM generates a valid solution. If one wants to use models provided through API like GPT-4 from OpenAI, the latency of the models will depend on a list of factors like network connection and server availability. In our

paper, we typically need to wait around 15-30 seconds for one step to get completed, where less than 0.1 seconds are used for local processing.

## 5 Related Works

**LLM for Reasoning**  Since the release of GPT-3, researchers have studied the effect of how they are performing on diverse sets of problems. Ideas like chain of thought Wei et al. (2022), in context learning Ye et al. (2023) have been shown to significantly improve the performance of LLMs in different tasks related to reasoningFu et al. (2022); Shum et al. (2023). A recent line of work has been introduced to use natural language as feedback in the process to introduce iterations and give LLMs more than one chance to generate correct solutions, which has been shown to significantly help in code generation and reasoning Chen et al. (2023a); Yang et al. (2023); Shinn et al. (2023). Specifically in LLMs for planning, there is a line of work that shows how bad LLMs are in general planning domains Valmeekam et al. (2023; 2024a), some works on benchmarking the performance of LLMs on task planning or travelplanning, which while share the name of planning, focus more on reasoning Zheng et al. (2024); Xie et al. (2024), and a few papers on combining the LLMs with search and evolutionary algorithms and improve the performance of LLMs on reasoning problems, especially on planning problems where search are traditionally performing well Chen et al. (2024); Su et al. (2024); Lee et al. (2025). In this paper, we are specifically looking into a specific problem in planning.

**LLM for Multi-agent Systems**  While there is extensive research on using LLMs for many different problems, there is little work that addresses the problem of LLM for multi-agent systems in the beginning. As time proceeds, there has been a list of works that promote research in social behavior by creating multi-agent environment powered by a lot of LLM agents Tan et al. (2023), create dialogue-based games with LLM Schlangen (2023). Recent work has shown that dialogue between multiple agents can help remove factual errors Du et al. (2023); Wang et al. (2023b). When it comes to solving problems related to multi-agent system, there are some works that are related to robotics Zhang et al. (2023); Mandi et al. (2023) while limited to the scale of two to three agents. Chen et al. (2023b) studied whether using a more decentralized controller can help to solve the context length limit problem when the number of agents is higher, and concluded that in a centralized controller environment, creating separate LLMs for different agents does not help improve the success rate. While they succeed in their domains, in this paper, we specifically look at the problem of MAPF, and we show the cause of failure in the MAPF problem because of its unique challenges.

**Multi-agent Path Finding**  Multi-agent path finding (MAPF) is a problem that has won much attention in recent years because of its close relationship to real-world applications. It has previously been solved with more classic methods like heuristic search algorithms Sharon et al. (2015); Li et al. (2021); Okumura (2023), rule-based algorithms Han & Yu (2020), and reduction-based algorithms Surynek et al. (2016). While learning-based approaches have not yet outperformed classic approaches, they have also begun to win a lot more focus for their fast inference time and generalizability. PRIMAL Sartoretti et al. (2019) proposed to learn a policy for MAPF using a combination of reinforcement learning and imitation learning. Following that, a group of works proposed a diverse set of methods from building curriculum Damani et al. (2021) to follow guidance from classic methods in each step Skrynnik et al. (2023). In this paper, we focus on using LLM, which is a learning-based approach but is not specifically fine-tuned for MAPF problems.

## 6 Conclusion

In this work, we introduce the Multi-Agent Path Finding (MAPF) problem as a novel and structured benchmark for evaluating the reasoning capabilities of large language models (LLMs). We present a framework for formulating MAPF as a benchmark task and empirically demonstrate that, while LLMs can solve simple MAPF instances, their performance degrades significantly as problem complexity increases. To better understand the specific challenges posed by this benchmark, we analyze the failure modes of LLMs and categorize them into three distinct capability gaps, namely reasoning capability, long-context capability, and understanding symbolic information. Through extensive experiments, we validate this categorization and

discuss how addressing these challenges may require insights from diverse research communities, including planning, representation learning, and natural language understanding.

## Acknowledgements

The National Science Foundation (NSF) supported the research under grant number 2112533: "NSF Artificial Intelligence Research Institute for Advances in Optimization (AI4OPT)".

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

## A  Complete Prompts

You are a decision-maker in a warehouse system that is running as a 4-connected grid-based multi-agent path finding system of [[Map Description]]. You will be provided with the current scenario in which you need to choose actions for each agent, i.e., the detailed information that provides where the obstacles are, together with each agent's current location and goal locations. The objective is to minimize the time of the agent who arrives at its goal location at the latest while keeping no agents from colliding with each other at any timestep. You only need to consider collision between two agents located in the same cell after each move. On each timestep, you could choose one action from moving up, moving down, moving left, moving right, and staying without moving. You should try to avoid collisions not only in the current timestep but also look ahead to a few timesteps if it is important to avoid future collisions.

Remember that to avoid a collision, normally, only one agent needs to change their action from the original action. Nevertheless, your job is not to generate a whole path but to do it step-by-step, and the information on whether a specific action is valid or not is provided in each single step. Remember that the given action is based on the obstacles, so the available action will not change if the agent chooses to stay at the same location. You only need to give a single-step action for each agent to follow; do not give more steps until I tell you to.

First, give your solution in the same format as follows, with some explanation if the agent is not moving or moving in a direction that is not towards its goal. (This is not the actual scenario, but an example of the desired output format. You will be provided with the actual scenario later):

Step 1:
Agent 1:
Current: (1,0)
Goal: (1,0)
Move: Stay, as it has already reached its goal.

Agent 2:
Current: (5,4)
Goal: (5,6)
Move: Up to (5,5)

Agent 3:
Current: (0,7)
Goal: (6,4)
Move: Right to (1,7)

Then, validate your solution by listing all the coordinates of the agents after the move and check if there is any pair of agents that have the same coordinates. If there is a collision by mistake, regenerate another solution immediately and validate again by listing the coordinates of all the agents until you get a collision-free solution. Finally, end your output with your validated solution in a new paragraph with the format of a sequence of : [[Agent_id]]: [[Coordinate]].

Figure 6: An example of the system prompt to specify the MAPF system and the objective. The text highlight in blue will be replaced by map description. For example, in room-32-32-4, the prompt will be 'room-like map with size 32*32'.

In Fig. 6, we give our system prompt. Between different maps, the only thing that changed is the text highlighted in blue, and all other parts are the same. We provide the user prompts in Fig. 7 and Fig. 8.



Agent 1 is currently in (0,2), and wants to go to (3,1).
Agent 2 is currently in (1,3), and wants to go to (2,0).
The map is as follows, where '@' denotes a cell with an obstacle that an agent cannot pass, and '.' denotes an empty cell that an agent can pass.
The bottom-left cell is (0,0) and the bottom-right cell is (31,0):
....
...@
....
.@..
In the next step:
Agent 1 can move ['stay at (0, 2)', 'right to (1, 2)', 'up to (0, 3)', 'down to (0, 1)'].
Agent 2 can move ['stay at (1, 3)', 'left to (0, 3)', 'right to (2, 3)', 'down to (1, 2)'].



Figure 7: An example of the user prompt for describing the scenario. Text in blue is a scenario-specific prompt, while text in orange is a map-specific prompt. In the experiments on the empty map, only the first blue paragraph will be provided, and all text starting from the black paragraph is removed because there are no obstacles. The text in purple is the single-step observation (SSO) information.



[[Success]]
Good job. Keep moving. In the next step:
Agent 1 can move ['stay at (0, 2)', 'right to (1, 2)', 'up to (0, 3)', 'down to (0, 1)'].
Agent 2 can move ['stay at (1, 3)', 'left to (0, 3)', 'right to (2, 3)', 'down to (1, 2)'].
[[Failure]]
You are wrong. Agents (1,2), and (4,5) are colliding with each other. Please correct the current step.
You are wrong. Agent 2,4 is colliding with obstacles. Please correct the current step.



Figure 8: An example of the user prompt starting from the second step. While here we demonstrate a few options, only one of them, i.e., text in one color, will be provided to the LLM in one iteration.

## B  Token Length Grows

In Fig. 9, we show the average prompt length per agent per step, which includes both the input and the output. We found that with the growing number of agents, this average is also growing, which means that the output length is growing faster than linear. This contributes to the fact that the more agents, the more complex the environment, and thus, the longer the explanation and the more iterations until a correct plan is found. We also observe that in failing scenarios where the agents go back and forth, the token length is generally shorter, given that they did not run into the iterative deconflict process, which takes a lot of tokens to solve.

## C  Comparison Between Rule-based Checker and LLM as Checker

In many domains, people are using LLM as the checker and provide feedback to another LLM, which serves as an actor. While we choose to use a rule-based checker because of its reliability and speed, here we provide some results on how bad the performance of an LLM-based checker could be. We randomly picked 50 different steps of information that need the checker to verify from our discussion history. And the success rate is 76%, and the average time spent for each check is 3 seconds. This success rate is why we choose to use the rule-based checker.

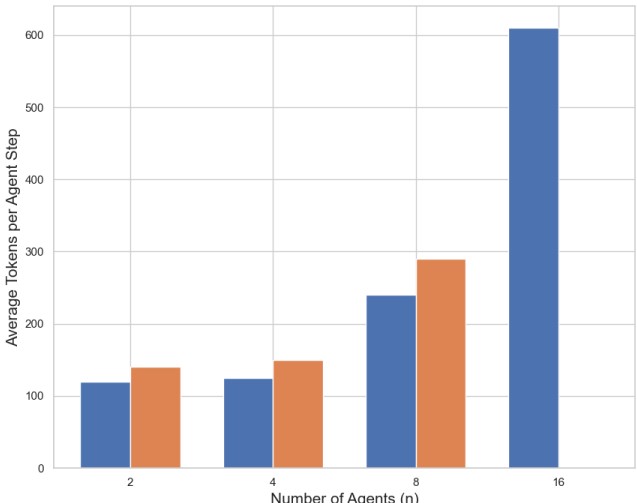

Figure 9: Average token length per agent step on success scenarios.

# D    Complete Failing Example

While due to space limits, we have only provided part of the example of the symmetry breaking examples in the main paper, here we provide the full example in Fig. 10 on the corresponding step as an example of what will happen in one step of LLM response.

# E    Case Study on GPT-o1-preview

In the main paper, we showed that the model of (GPT-)o1-preview has a worse performance than the previous GPT-4-turbo. To study this anomaly, we provide a case study here.

As shown in Fig. 11, compared to the previous version, GPT-o1-preview now learns to explicitly try to find an alternative route, and also, it now learns to use its previous history more strategically. After the model has found out that moving right is probably not helpful to him, it will not try to go right again. However, this is making the result worse, given the current capability of the model is not strong enough. We observe that while at location (1, 17), agent 5 could have moved up, GPT-o1-preview still chose to go back to the left. This fails to meet the motivation of trying to find an alternative path when meeting an obstacle. It is noteworthy that after a retry, it chooses to move up and end up in a successful final plan. Similar scenarios happens which leads to the improvement in Table. 11, which shows that GPT-o1-preview has the capability of at least matching the performance of the previous model, but the current result is coming from the randomness of LLMs. Also, this opens up the potential for future research to study how to automatically integrate the regeneration process when provided with a more advanced checker. This checker would need to go beyond merely assessing the final output, like the current checker, and instead evaluate the entire reasoning process. The checker should be able to predict whether the current explanations are likely to lead to failure in future steps, rather than simply determining if the current step is correct. This aligns with evaluating the $Q^\pi$ value in a Markov decision process, where $\pi$ represents the generation policy of the large language model (LLM). Unlike $V^*$, which evaluates the current state based on an optimal policy, $Q^\pi$ would allow the model to assess whether the current action will result in long-term success or failure in tasks such as MAPF. For instance, the checker should recognize that while the agent could theoretically move back to (1, 17) later, if it chooses to move left to (0, 17) at step 6, the model's policy will not opt for this, indicating the need for regeneration at this step rather than later. It is also noteworthy that even when large reasoning models (LRM) like GPT-o1-preview have success at some point, it will also introduce a much

Step 1:
Agent 1:
Current: (0,2)
Goal: (3,1)
Move: Right to (1,2)

Agent 2:
Current: (1,3)
Goal: (2,0)
Move: Down to (1,2)

After the move, the coordinates of the agents are:
Agent 1: (1,2)
Agent 2: (1,2)

There is a collision between Agent 1 and 2. I will regenerate another solution immediately.

Step 1 (Regenerated):
Agent 1:
Current: (0,2)
Goal: (3,1)
Move: Down to (0,1)

Agent 2:
Current: (1,3)
Goal: (2,0)
Move: Left to (0,3)

After the move, the coordinates of the agents are:
Agent 1: (0,1)
Agent 2: (0,3)

There is no collision. Here is the validated solution:

Agent 1: (0,1)
Agent 2: (0,3)

Figure 10: The output from GPT-4-turbo for the symmetry breaking example in Fig. 3b.

longer inference time and make it even harder to deploy in the real world. We also provide another example in Fig. 12, where GPT-o1-preview has also shown its reasoning of trying one step, and going back. However, this example goes back directly because it sees no option to go to the goal location after the detour. This is one of the examples that LLM still fails even after we incorporate the human checker in the middle because it still does not have a strong capability in path planning.

<Assistant>
Step 5:
...
Agent 5:
Current: (0,17)
Goal: (0,23)
Move: Right to (1,17), aiming to find a path upwards since moving up is blocked.
...
<User>
Keep moving. In the next step, ...
<Assistant>
...
*Charting new routes...*
*Agent 5 is mapping out new paths towards (0,23). Moving left or up from (1,17) could be viable, but obstacles might alter their course. Interessen her own challenges and adjusting trajectory.*
...
Step 6:
...
Agent 5:
Current: (1,17)
Goal: (0,23)
Move: Left to (0,17)
...
<User>
Keep moving. In the next step, ...
<Assistant>
Step 7:
...
Agent 5:
Current: (0,17)
Goal: (0,23)
Move: Stay at (0,17), as moving right doesn't help and moving up is blocked.
...

Figure 11: An example of part of the output from GPT-o1-preview model. The main difference compared to previous model in the reasoning is highlighted in red. Italics are the part shown as the thought of GPT-o1-preview in the web version.

<Assistant>
Step 8:
...
Agent 3:
Current: (17,5)
Goal: (17,1)
Move: Up to (17,6)
Explanation: Moving up to find a path that allows decreasing the y-coordinate towards the goal.
...
<User>
Keep moving. In the next step, ...
<Assistant>
Step 9:
...
Agent 3:
Current: (17,6)
Goal: (17,1)
Available moves: ['stay at (17,6)', 'right to (18,6)', 'up to (17,7)', 'down to (17,5)']
Move: Down to (17,5)
...
<User>
Keep moving. In the next step, ...
<Assistant>
Step 10:
...
Agent 3:
Current: (17,5)
Goal: (17,1)
Available moves: ['stay at (17,5)', 'left to (16,5)', 'right to (18,5)', 'up to (17,6)']
Move: Stay at (17,5)
Explanation: No available moves bring Agent 3 closer to its goal without moving away.
...

Figure 12: Another example of part of the output from GPT-o1-preview model.

