# OpenReview forum: "Solving Multi-agent Path Finding as an LLM Benchmark: How, How Good and Why"
_TMLR — Accepted by TMLR_

### Review · Reviewer_FMAf · 2025-02-25

**Summary Of Contributions:**

1. This paper proposes using an LLM to directly solve MAPF, providing a new benchmark to evaluate LLMs' planning and spatial reasoning capabilities.

2. Three causes of failure are presented and analyzed.

**Audience:**

Yes

**Claims And Evidence:**

Yes

**Requested Changes:**

1. In the single-agent scenario, only a simple prompt is used. It is unclear whether an advanced prompt (e.g., verbal RL [Ref A]) would improve performance.

2. In the multi-agent scenario, the prompt may become long and reach the context limit. Is it possible to model each agent with one LLM and enable communication among LLMs (e.g., [Ref B])? Will this alleviate the issue?

3. It is unclear if an advanced spatial reasoning prompt (e.g., [Ref C]) will help in this task. Could it improve the understanding of obstacles?

4. Multi-agent pathfinding can be solved by CBS, and GPT is familiar with this algorithm. How will GPT behave if we prompt it to solve the problem using CBS?

I strongly encourage the authors to include the methods mentioned above and conduct ablation studies.

5. Compared with traditional spatial reasoning, the multi-agent setting is novel. The paper could benefit from a discussion on the difficulties GPT faces in understanding multi-agent scenarios compared with single-agent scenarios. When the number of agents increases, what challenges arise beyond the context length limit?

[Ref A] Reflexion: Language Agents with Verbal Reinforcement Learning

[Ref B] CAMEL: Communicative Agents for "Mind" Exploration of Large Language Model Society

[Ref C] Mind’s Eye of LLMs: Visualization-of-Thought Elicit Spatial Reasoning in Large Language Mode

**Strengths And Weaknesses:**

**Strengths**

1. The proposed benchmark is novel and will inspire future works to evaluate LLMs' planning capabilities, especially in a multi-agent scenario.

2. The provided causes might be helpful for other planning and spatial reasoning benchmarks.

**Weaknesses**

1. Only simple prompts are evaluated; it is unclear whether advanced prompts can drastically improve performance. (Please refer to requested changes 1-4)

2. The three causes have been discussed for general planning problems in the literature. It seems that there are no unique insights specific to multi-agent or path planning problems. The unique insights beyond these general points should be discussed. For example, the requested changes. I hope the authors can discuss other unique insights. (Please refer to requested changes 50)

3. (Multi-agent) pathfinding belongs to the category of spatial reasoning in LLM literature. More relevant literature should be discussed.

---

> ### Author Response · Authors · 2025-03-14
>
> Thank you for your detailed suggestions. First, we invite you to review our general responses regarding the motivation behind our work, as they address some of the weaknesses you have highlighted.
> We also acknowledge that ongoing and future research in this field may introduce new approaches that could be beneficial to our setting. In response to Reviewer Fbb4’s suggestion, we have explicitly recognized this in our paper. Additionally, we have added a paragraph in the discussion section to include some of our previously unsuccessful trials.
> With regard to the specific requested changes:
>
>
> 1. While we acknowledge that more advanced prompting techniques may have the potential to improve performance, we respectfully disagree that verbal RL would be effective in this case. Reflexion-style prompting relies on self-evaluation and self-reflection to enhance performance. First, we have previously experimented with ReAct and observed no improvement when applied to GPT-4-turbo. Since Reflexion builds upon ReAct, we do not find it promising in this context. Additionally, the latest o1 models already incorporate self-reflection as a built-in capability, yet, as demonstrated in our paper, they do not exhibit significant improvements on this task. Given these findings, we do not believe evaluating Reflexion would add meaningful insights to our study, but we are open to include any other techniques that you feel promising.
> 2. Indeed, decentralized can be a useful way to mitigate the context-length limits. However, Chen et al. [1] have shown that decentralized control is not significantly superior on performance compared to a centralized system. (note that their proposed method of HMAS is, in fact, a special case of our workflow where the centralized system is replaced by the rule-based checker.) To reduce duplicity in the field, we have included them in our related work rather than comparing the similar decentralized methos again.
> 3. Thank you for the suggestion. We have been conducting experiments on this with GPT-4o due to the retirement of earlier model we used in our paper. And we found that even if with this model, the success rate is still 20% with GO+SSO in Table 4 in our on the room-32-32-4 map with 8 agents, which does not show any advancement. We have included the method in the discussion section as an additional failing trial.
> 4. We agree that this is a promising direction. However, we believe that it deviates from our objective to measure the capability to solve the problem by LLM itself without guidance from humans. We have included a similar statement at the end of Sec. 3.2.
> 5. While the three factors analyzed in this paper also apply to single-agent planning, the multi-agent setting introduces additional challenges due to the output length increasing at least linearly with the number of agents. Longer outputs expand the total context length, necessitating a restart mechanism in our algorithm. This mechanism reinitializes the entire LLM system with a new problem, using the current locations of all agents as their starting points when context length limits are reached. While this approach addresses the immediate problem, it negatively impacts final solutions by causing the algorithm to lose earlier information, such as each agent's preferred direction and potential map locations that the LLM initially struggled to encode. These losses further exacerbate the other two challenges discussed.
>
>      In single-agent settings, the LLM's extended input context can help avoid repeated paths, even when the model does not perfectly understand obstacle locations. Similarly, the solution checker can guide the LLM in producing viable paths even if the generated path is suboptimal. However, in multi-agent settings, the limited capabilities of LLMs lead to more frequent mistakes, resulting in additional restarts. Each restart compounds the loss of historical information, such as agent preferences and map details, ultimately increasing failure rates.
>
>      Our experiments demonstrate that on an empty map, where collision avoidance is the only constraint, LLM solvers can effectively scale to handle up to 16 agents (one agent for every four cells on the map). However, on larger maps with fixed obstacles, LLM solvers struggle even with only eight agents. Comparing these results suggests that agent collisions are not the primary factor driving the performance gap. Instead, the increased output length appears to be the main reason for the performance differences between single-agent pathfinding and multi-agent pathfinding (MAPF).
>
> [1].  Chen, Y., Arkin, J., Zhang, Y., Roy, N., & Fan, C. (2024, May). Scalable multi-robot collaboration with large language models: Centralized or decentralized systems?. In 2024 IEEE International Conference on Robotics and Automation (ICRA) (pp. 4311-4317). IEEE.

---

> > ### Comment · Reviewer_FMAf · 2025-03-31
> >
> > Thanks for the detailed response. All of my concerns have been solved.

---

### Review · Reviewer_TpMx · 2025-03-06

**Summary Of Contributions:**

The authors focus on the task of solving traditional MAPF problems with LLMs through various prompting techniques, and reporting on the results that they find. The authors found that state-of-the-art LLMs are capable of producing solutions for the easiest of MAPF tasks (e.g., single agent in rooms without obstacles), but fail to generalize to anything of reasonable size.

There are three issues raised by the authors as potential sources for the poor performance: (1) lack of LLM understanding; (2) limits on the context length; and (3) lack of reasoning capability.

Finally, the evaluations to demonstrate the behaviour cover a range of LLM models, prompt styles, and benchmarks.

**Audience:**

No

**Broader Impact Concerns:**

I don’t feel that there are any broader impacts or ethical concerns that the authors need to address with this line of work.

**Claims And Evidence:**

No

**Requested Changes:**

I have a hard time knowing what to suggest, as my core criticism with the work isn’t just something a revision can fix. I suppose that one thing I might recommend to the authors is to take the insight and expertise they clearly have in the MAPF field, and apply it to novel / meaningful settings that include LLMs. Can they find suboptimal parts of solutions? Can they explain existing MAPF plans? Can they be used to synthesize programs for searching or heuristics? Etc. These are avenues in the same ballpark that don’t rely on winning the horse race against well-established and efficient methods.

Aside from that major suggestion, these are the minor things I noticed while reading the paper (as mentioned previously, it was otherwise extremely well written).

## Minor fixes

- There are some references that need fixing. The ones I’ve noticed include…
  - First sentence of 2.3.2
  - Question mark at the bottom of pg 9

- Start of the intro needs to be updated — it’s been more than a year since ChatGPT has come out.

- Not sure what you meant by “overcooked” on the first page

- At the end of 2.2, you should explain what is meant by “getting a rate limit error”. I think I know what you mean (being restricted by the API call limits on the LLM service you use), but this should be elaborated on.

- The use of “Besides, …” to start a sentence (twice in a row on pg 9) is too informal.

- Not sure what “stably” means in the second sentence of the Conclusion section.

**Strengths And Weaknesses:**

In general I found the paper to be well written (minor fixes on wording and such are listed below), but fundamentally, the paper explores what I feel to be a fairly inappropriate line of exploration. It should be obvious that LLMs cannot do the long-sequence reasoning required for complicated MAPF problems, and unless this is surprisingly not the case, I fail to see the merit in running evaluations to confirm this hypothesis. To put it analogously, I don’t think LLMs are capable of computing the mean of a large set of floating-point numbers. It would be surprising and worth exploring if this were not the case, but I don’t believe a paper is warranted just to establish that LLMs failing at large-scale computation of the mean.

The title of the paper is provocative: ``_Why_ Solving MAPF with LLMs has not Succeeded Yet’’. Such a bold statement requires similarly bold evidence, but I found the three reasons provided (cf. Section 3) to be largely surface level and kind of obvious. LLMs are doing token prediction and not reasoning, and it’s obvious that tasks dedicated to complicated reasoning would be something that they struggle to handle. This all seems to step from a position of “hoping LLMs work” for this setting (cf last sentence on pages 2 and 4), but the very premise doesn’t make sense. If we are in such a structured environment where collisions are trivially detectable, then why would we ever turn to a large set of weights to only maybe compute the right answer part of the time? “Current heuristic-based algorithms like CBS can solve all the scenarios tested in the paper in less than 0.1 second” <— this should be a clear indication that turning to LLMs for solutions is a losing prospect.

The authors have clearly explored a wide variety of angles to try and have LLMs auto-complete their way to a MAPF solution. It’s definitely a positive sign that multiple things were attempted (e.g., range of maze types/size and the different input forms found in Figure 5). And as far as I can tell, the evaluations across these options are done correctly. The details included in the appendix are also nicely thorough.

However, I ultimately feel that the motivation of the work is on shaky ground, and it’s not something the readership of TMLR would be compelled to know about.

---

> ### Author Response · Authors · 2025-03-14
>
> We appreciate the reviewer’s feedback and have addressed the minor fixes mentioned. We invite you to review the revised version of our paper.
>
> Regarding your core criticism on the motivation for this work, we argue that MAPF is comparable to well-established benchmarks such as BlocksWorld and the more general PlanBench. These benchmarks have been instrumental in advancing research, and we believe MAPF holds similar potential as an interesting and solvable problem for the community to explore. Additionally, even in widely used mathematical reasoning datasets like GSM8K and MATH—both of which are relevant to your argument regarding “I don’t think LLMs are capable of computing the mean of a large set of floating-point numbers”—the initial assumption was that LLMs would struggle with such tasks. However, as these benchmarks gained exposure within the LLM community, performance on them improved significantly. We believe that these popular reasoning-based datasets collectively support the motivation behind our work: introducing another benchmark where LLMs have not yet demonstrated strong performance. Our paper aims to help the broader LLM research community understand the specific challenges MAPF presents, particularly in terms of long-context limitations, reasoning capabilities, and the ability to process symbolic representations. Therefore, we believe our paper should be an interest to the TMLR community, which is supported by other reviewers.

---

> > ### Comment · Reviewer_TpMx · 2025-03-19
> > **Rebuttal Received**
> >
> > Thank you for your response on this. I understand the motivation, and appreciate the fixes to the minor issues (raised by all reviewers) that you have put into the work. However, I remain unconvinced that MAPF as a setting provides any insight beyond works such as PlanBench to demonstrate a situation where LLMs are inherently incapable of solving. It’s long-sequence reasoning that have hard constraints that must be satisfied (and, indeed, MAPF can be recast as a planning problem and the insights/reasoning behind LLMs failing would remain).
> >
> > My view on the line of work hasn’t changed as a result of the rebuttal, but would continue to encourage the authors to find other avenues within the MAPF space where LLMs can offer a unique and compelling contribution.

---

> > > ### Author Response · Authors · 2025-03-19
> > > **Further comments**
> > >
> > > Thank you for your further response. We would like to respectfully reiterate why we believe our current framework for applying LLMs to MAPF is valuable to the TMLR community.
> > >
> > > Most importantly, as LLMs continue to advance rapidly, there is an increasing need for diverse benchmarks to evaluate their performance, particularly in reasoning tasks. This is why numerous new datasets have recently been introduced, such as mathematical reasoning benchmarks like AIME, code-based benchmarks like MBPP, planning-related benchmarks like PlanBench, and more general LLM-agent datasets like Travel Planning and NaturalPlan. Many of these datasets align with your observation that “LLMs are inherently incapable of solving” certain tasks. However, the primary purpose of such benchmarks is not to provide immediate solutions but rather to serve two key roles: (1) as potential future training resources to enhance LLM reasoning capabilities, and (2) as evaluation tools to assess whether LLMs are making meaningful progress toward more general intelligence.
> > >
> > > The MAPF problem introduced in our paper is uniquely positioned as a benchmark, incorporating both planning and multi-agent cooperation. Moreover, it is inherently adaptable in difficulty by varying the number of agents involved, making it a promising candidate for future benchmarking. Our primary goal is to establish MAPF as an interesting and valuable benchmark, rather than to develop an alternative solver for MAPF. This is similar to how PlanBench does not aim to replace well-established solvers based on the Planning Domain Definition Language (PDDL) but instead focuses on evaluating LLMs’ planning capabilities. While there may be other promising avenues for MAPF research, our proposed framework is unique and provides a meaningful benchmark for evaluating LLMs. And as the first work to explore the use of LLMs in MAPF, we also believe this integration paves the way for future advancements, such as language-based instruction for MAPF.
> > >
> > > Overall, given the purpose of our paper, we believe our work is important in introducing MAPF to the broader LLM research community, which is a key subgroup within TMLR.

---

### Review · Reviewer_xdjV · 2025-03-08

**Summary Of Contributions:**

This article examines the ability of current LLMs to directly produce plans for single-agent and multi-agent pathfinding problems. It uses a set of publicly available four-connected grid-based pathfinding problems, and uses a number of different LLM arrangements to navigate to the goal location through the use of prompts and answers. The authors give experimental results showing an LLM setup can solve simple pathfinding problems, but the success rate for a variety of setups decreases to total failure as the pathfinding difficulty increases.. The authors give some analysis looking at different types of problem difficulty, prompts, multimodal input, different LLMs, and the effects and constraints of context length.

**Audience:**

Yes

**Claims And Evidence:**

No

**Requested Changes:**

Section 1
“MAPF is a unique problem in multi-agent coordination, so it is challenging …”
Authors should expand on what is unique. For example, what is a notable difference from cooperative multiplayer games?
Also, challenging doesn’t follow from uniqueness. This sentence needs some care and re-wording.

“On the other hand, path planning is one of the easiest parts of planning”
easiest in what context and in what sense? what are the other parts?

Section 2.2
“Instead, it breaks down the whole planning task into smaller singler-step tasks”
The authors do also explore a setup more like chain-or-thoughts (rhe one-shot setup). Rather than saying SBS is different, this paragraph would be a good spot to note the authors look at both. OS should be introduced here regardless, as it shows up soon after in the Table 1 results. The paragraph in 2.3 “It has been shown that a step-by-step generation is better in many reasoning-related tasks …” should probably be moved here, as part of the discussion of the general experimental setup.

Section 2.3
The results in Table 1 and Table 2 present the binary success criteria. This is reasonable for simplicity. However, less-processed results should also be included: what do the results look like in terms of solution length and single-step failures (replanning events)?

Figure 3: explain what the authors mean by symmetry breaking here: there are no other mentions of symmetry in the paper.
Collisions at (2,1) and (1,2)?  For agent 1/agent 2 the paths [R/D] and [D/R, R/D, R/D] both lead to collision, and at different locations. This is possibly related to the question of symmetry above, but as noted right above, there is no discussion of specific handling of rotations/flips.

“We observe that for smaller-scale problems that have fewer agents, GPT-4 can successfully generate valid solutions. However, GPT-4 fails to generate valid solutions”
Elsewhere, the authors have a distinction between valid and successfully generating a solution with few enough steps. So in these cases here, do the author mean valid, or successful within the 3*optimal condition?

Section 3.1
“ in scenarios that are successfully solved, this average is 1.5”.
What is the ratio in the problems where the LLM agent is not successful?

“LLMs can mostly succeed in scenarios that do not need a lot of waiting, and in most steps, they only need to go in the two directions aligned with the goal’s direction”
The clause about direction is separate from and does not follow from the ratio argument. It should separately justified with data.

“This lack of reasoning capabilities is usually solved with tool use in other domains, but MAPF itself is hard because MAPF requires the capability of path finding in a complex map and avoiding collisions between all pairs of agents strategically and efficiently.  …”
This paragraph is in my opinion making an unjustifiably broad claim, and I would suggest weakening it. The most obvious way to apply a tool doesn't work, but that doesn't necessarily mean that the hardness is a fundamental feature of the problem. For example, having a complete plan, in a similar way to SBS,  seems like it would allow for a non-solving collision-checking tool.

Table 3. Link GO and GO+SSO to the previous discussions. GO+SSO is what was used for Table 1 and 2?  If so, possibly consider renaming GO+SSO to be OS to match past experiments, and maybe GO to be OS-SSO?

Similar for Figure 5. How do TOM and TOO relate to previous experiments? IIt seems like they must be TOM: the correct answer should be obvious in the caption and text

— smaller issues —

Section 2.3.2 “Based on our results on single-agent path planning, we believe there is a possibility of generating a plan, regardless of optimality, for MAPF.”
I am unsure what the authors are trying to convey to the reader here: suggest either dropping it, or rephrasing it if the authors believe it is conveying something important.

Section 2.3.3 “While it is unclear how well LLMs can solve MAPF problems”
Why is it unclear at this point in the paper? One possibility is that this statement is obvious (and unnecessary) readers haven't done the experiments and the paper hasn't given results. Another possibility is that the authors believe the readers should be unsure at this point: if so, that wasn't clear to me, and the authors need to make that expected reason clear.

Section 3.1 “More specifically, LLMs are guaranteed to tell whether”
not guaranteed?

“We also observe that the GPT-4-8k model takes more iterations than the latest GPT-4-turbo
model. This could either contribute to forgetting earlier information or to the improved capacity of the new model.”
What is contributing to forgetting? Please rephrase the second sentence.

Section 4
“LLMs can solve small problems simply through prompting and discussing what is stopping them from solving larger scenarios”
Suggest re-phrasing this is a weaker statement, e.g. “... and discussing some difficulties LLM have when solving larger scenarios”.

“While we are unclear on how such things … will be a very challenging but meaningful direction… we want to remind them to keep looking …”
Suggest softening some of the wording here, using could instead of should, and phrasing in terms of the authos believing something to be a promising direction. There is something slightly awkward about saying "we are unclear on how", and then directing researchers in very specific directions. Or, as an alternative, just drop text from "Furthermore ..." onwards: there's no need to reach so hard for future work, it is mostly speculative.

**Strengths And Weaknesses:**

This paper is reporting a negative result. One possible positive outcome of a negative result is a surprise: within a class of problems where a method has usually worked, there is an unexpected instance where it does not. I’m not personally surprised an LLM is not directly capable of solving a maze, nor does the article clearly make that claim.

Another way a negative result can be useful is preventing other researchers from needlessly repeating the same experimental idea. This utility is proportional to how well the obvious ideas are covered, and how much interest there is in the problem. The authors do try a range of things, drawing from existing literature.

As the authors also point out, the lack of success could also be considered a motivation for researchers to consider this a challenging problem that may spur development.

However, those two benefits are in conflict with each other. If people are already (potentially) looking at this so that it’s worth presenting the results and preventing duplication, then it’s not clear people need to be motivated to look at the problem. But if this is a challenging problem that not enough people are looking at, then showing the obvious things don’t work is not as useful on its own. As things are written, the article does seem to be presenting both of these, and I think it would be stronger to argue for only one (and easier to justify those claims).

Additionally, while the authors did try a number of different variations, it’s not clear to me that it’s reasonable to consider the space of obvious solutions to be covered well, when LLMs are currently an extraordinarily active topic of research for a broad range of problems. This applies both to methods of using LLMs, through the discovery of more generally effective prompting schemes, as well as LLMs themselves, where drastic improvements in context length could solve some of the pathfinding failures the authors note might be (partially) caused by lack of history.

---

> ### Author Response · Authors · 2025-03-14
>
> We sincerely appreciate the reviewer’s detailed feedback. We have addressed all the requested changes, and we invite you to review the revised version of our paper.
>
> Regarding the weakness you pointed out, we completely agree that, as a benchmark paper, our goal is to encourage future researchers to engage with this benchmark without requiring them to replicate our exact approach. We encourage you to refer to our general responses on the motivation behind this work, and we would appreciate your feedback on whether those clarifications address your concerns.
>
> Additionally, regarding the coverage of methods, we acknowledge that our initial argument may have come across as too strong, potentially implying a more comprehensive coverage than intended. In response, we have incorporated Reviewer Fbb4’s suggestion to explicitly recognize the potential for new approaches and methodologies. Our intention is for this paper to serve as a foundational building block for future research, and we hope this revision makes that clearer.
>
> Thank you again for your thoughtful comments!

---

> > ### Comment · Reviewer_xdjV · 2025-03-21
> > **Comments after author response**
> >
> > Thank you to the authors for their response, and the updates in response to initial feedback
> >
> > However, I still have some trouble with the high-level motivation.
> > In order to follow the clarified position in the general response, that the paper is intended to be proposing multi-agent pathfinding as an LLM benchmark problem, I think the paper would still need a substantial re-write. Excluding some sentences added to the introduction and discussion section, the language – from the abstract, to the introduction, to the conclusion, and even the title – is limited to discussing the problem of why a number of LLM-based solution methods do not work for the domain, and not in proposing a benchmark problem. There is significantly more text spent describing the methods used, and their shortcomings, than describing the proposed benchmark problem, and trying to set it up as a benchmark that results in future experiments that can be compared to each other for a demonstration of progress.

---

> > > ### Author Response · Authors · 2025-03-21
> > > **Further Comments to Reviewer xdjV**
> > >
> > > Dear Reviewer xdjV,
> > >
> > > Thank you for your response. And thanks to your response, we realize that we might have caused some confusion in our global response as it has combined the questions from a few reviewers. So here, we want to clarify again:
> > >
> > > The primary motivation of our work is to introduce MAPF to the broader LLM community (as elaborated in our other global response) as a new benchmarking domain, which is why we believe this paper is relevant and beneficial to the TMLR community. The core position (claim) and contribution of the paper is to outline potential reasons for “Why solving MAPF with LLMs has not succeeded yet.” And to support it, the paper is focusing on evaluating multiple prompting strategies to support our proposed breakdown of the key challenges involved in LLMs’ weak performance on MAPF.
> > >
> > > We have also posted this clarification to the general response in case other reviewers have similar concerns. If you have any further questions or concerns or suggest changes, please let us know, and we would be happy to address them.
> > >
> > > Best,
> > > Authors

---

### Review · Reviewer_Fbb4 · 2025-03-09

**Summary Of Contributions:**

The study examines large language models (LLMs) in addressing multi-agent path finding (MAPF) challenges. LLMs show success in solving MAPF on empty maps but struggle with obstacles. Directly using LLMs for MAPF lacks success due to understanding limitations, context length, and reasoning challenges. Addressing these limitations and other real-world challenges is crucial for advancing LLMs in MAPF applications. The paper aims to guide future research in this area.

**Audience:**

Yes

**Claims And Evidence:**

Yes

**Requested Changes:**

In Section 2.3.2 Results and Section 3 Causes of Failures, certain phrases such as "unclear" and "not stated" could be rephrased to adopt a more constructive and research-oriented tone. Instead of stating that aspects are unclear, they could be described as requiring further clarification, open to interpretation, or presenting an opportunity for deeper analysis. Similarly, rather than indicating that something is not stated, it could be framed as yet to be explicitly defined, providing room for discussion or presenting an opportunity for future research.

These adjustments would enhance the readability of the paper and encourage further exploration rather than simply highlighting gaps. Adopting this approach would provide a more positive and forward-looking perspective for the reader.

Additionally, incorporating a comparison with existing MAPF plans could be considered for further analysis, though this is merely a suggestion rather than a recommendation.

**Strengths And Weaknesses:**

The authors discuss three key reasons why LLMs have not succeeded in solving MAPF problems:

Capability to understand the scenario: The authors find that LLMs cannot fully comprehend the MAPF scenario, especially the information about the obstacles in the environment. Providing the map information through text-only descriptions, either listing the obstacle coordinates or describing the map using symbols, does not help the LLMs understand the problem.

Context length limit: MAPF problems can involve a large number of agents, leading to long input sequences that exceed the context length limit of current LLMs. The authors observe that the number of tokens used grows non-linearly with the number of agents, and they have to restart the LLM frequently, causing it to forget previous information.

Reasoning capability: The authors identify that LLMs lack the necessary reasoning capability to effectively coordinate the movement of multiple agents and avoid collisions. While LLMs can follow simple instructions to move in a certain direction, they struggle with the complex reasoning required to deconflict the paths of all agents.

The authors discuss potential research directions to address these limitations, such as improving the capability of LLMs to handle long contexts, enhancing their reasoning abilities, and leveraging multimodal inputs that combine language and image information to better understand the MAPF scenario.

Additionally, the authors highlight other challenges in using LLMs for MAPF in real-world applications, such as the need for solution quality guarantees and the high latency associated with LLM-based approaches. They suggest that researchers from different backgrounds, including classic planning methods and reinforcement learning, could contribute to advancing the use of LLMs for MAPF.

In conclusion, the paper presents a comprehensive investigation into the limitations of using LLMs to solve MAPF problems and provides a detailed breakdown of the key challenges that need to be addressed. The authors hope their work can serve as a building block for future research in applying foundation models to the domain of multi-agent planning.

Weaknesses
To strengthen this section, I suggest further elaboration on this point by acknowledging that while the evaluation focuses on simple prompts, exploring the impact of more advanced prompts could provide valuable insights into the system’s scalability and performance potential.

---

> ### Author Response · Authors · 2025-03-14
>
> Thank you for your valuable and detailed feedback, as well as your support of our work. We have incorporated the requested revisions to better encourage future research in this area. Additionally, we remain open to addressing any remaining concerns if certain aspects of our position still come across as too firm.
>
> Regarding the comparison with existing MAPF plans, we would appreciate further clarification on the specific aspects you believe the comparison would strengthen our claims. We are happy to elaborate on this point if you could provide more details on what metrics would be most relevant.

---

### Author Response · Authors · 2025-03-14
**General Responses to Reviewers**

We sincerely appreciate the reviewers’ detailed feedback. In addition to providing individual responses, we have updated the manuscript to address the majority of your comments, with all modifications highlighted in blue. We encourage you to refer to the revised version for a clearer view of these improvements.

Several reviewers have raised concerns regarding the motivation behind our work. To clarify, we position MAPF as a benchmark similar to some more popular datasets such as BlocksWorld and the broader PlanBench. These benchmarks have been instrumental in advancing research, and we believe MAPF holds similar potential as an interesting and solvable problem for the community to explore. However, despite its significance, MAPF has received limited attention, and as demonstrated in our paper, LLMs have yet to achieve success in this domain. Given this unique gap, our goal is to introduce MAPF to a broader audience and establish it as a meaningful benchmark. To facilitate this, we have carefully broken down its specific challenges to provide a clearer understanding of what kind of capabilities the problem of MAPF primarily measures.

We have incorporated these clarifications into the revised manuscript and would appreciate any further feedback on whether these revisions adequately address your concerns.

Thank you again for your time and valuable insights!

---

### Author Response · Authors · 2025-03-21
**Further General Response to Reviewers**

We thank all reviewers again for the detailed reviews and the responses to our rebuttal. Through interaction with Reviewer xdjV, we realize that our initial global response may have caused some confusion regarding the position of our paper. We would like to take this opportunity to further clarify:

The primary motivation of our work is to introduce MAPF to the broader LLM community (as elaborated in our other global response) as a new benchmarking domain, which is why we believe this paper is relevant and beneficial to the TMLR community. The core position (claim),
 and contribution of the paper is to outline potential reasons for “Why solving MAPF with LLMs has not succeeded yet.” And to support it, the paper is focusing on evaluating multiple prompting strategies to support our proposed breakdown of the key challenges involved in LLMs’ weak performance on MAPF.

If there are any further questions or concerns, we are happy to answer them.

---

### Decision · Action_Editor_o1p3 · 2025-04-22

**Recommendation:** Accept with minor revision

**Comment:**

The submission elicited a clear split among reviewers. Reviewers FMAf and Fbb4 praised the paper’s comprehensive experimental regimen and its systematic identification of why off-the-shelf LLM prompting fails on multi-agent pathfinding (MAPF). They welcomed the work’s potential to seed a new planning benchmark and felt their constructive feedback—on tone and minor analyses—was satisfactorily addressed. In contrast, Reviewers xdjV and TpMx found the negative results largely unsurprising given known LLM limitations and argued that the manuscript oscillates between diagnosing solver failures and proposing MAPF as a benchmark without sufficiently developing the latter narrative.

Despite these divergent views, the empirical methodology is solid, and the failure-mode breakdown will benefit the LLM evaluation community. At the same time, the benchmark proposition needs sharper articulation and balance. Accordingly, we recommend Accept with Minor Revision, with the following concrete edits to unify the paper’s framing and amplify its community impact:
- Sharpen Benchmark Framing
    - In the Abstract, Introduction, and Conclusion, explicitly state that the primary contribution is to establish MAPF as a standardized LLM evaluation domain—analogous to BlocksWorld and PlanBench—rather than to propose new solver algorithms.
    - Clarify in the title or subtitle that this is a “benchmark proposal with failure-mode analysis.”
-Reorganize Narrative Balance
    - Streamline the detailed failure-mode sections (Sections 2–3) by condensing repetitive solver-failure anecdotes and relocating some examples to an appendix.
    - Elevate a dedicated “Benchmark Protocol” subsection that outlines problem instances, evaluation metrics, and minimal dataset splits for community adoption.
- Consistency of Motivation and Claims
    - Ensure the motivation statements consistently emphasize benchmark goals rather than only empirical “why LLMs fail.”
    - Reconcile any instances where the manuscript vacillates between “discovering failure causes” and “proposing future benchmarks.”
- Minor Tone and Wording Tweaks
    - Adopt more constructive phrasing (e.g., “presents an opportunity for deeper analysis” vs. “unclear”).
    - Soften absolute claims and incorporate the suggested language edits from Reviewer Fbb4.

**Audience:**

Yes. TMLR readers working on LLM evaluation, reasoning benchmarks, and planning will find the detailed negative results and analysis of failure modes useful. The paper surfaces concrete challenges (e.g., exponential token growth with agent count, collision‑induced symmetries) that inform both benchmark design and future model improvements. However, for readers seeking novel solver algorithms or surprising theoretical insights, the work may feel unsurprising, as some reviewers noted that it reiterates known LLM limitations (long‑sequence reasoning, token‑based planning).

**Claims And Evidence:**

The paper’s core empirical claim—that modern LLMs systematically fail on multi‑agent pathfinding (MAPF) due to limited environment understanding, context‑length constraints, and reasoning shortcomings—is well supported by a comprehensive suite of experiments across different prompt styles, LLM variants, and problem complexities. The authors provide clear success/failure rates, ablations on single‑ vs. multi‑agent settings, and illustrative failure cases (collision hotspots, restart costs) that convincingly demonstrate why off‑the‑shelf LLM prompting does not solve MAPF. However, the secondary claim of establishing MAPF as a new, well‑justified benchmarking domain is less convincingly argued. While the authors compare to existing benchmarks like BlocksWorld and PlanBench in their responses, the manuscript itself devotes more space to “why LLMs fail” than to laying out a standardized benchmark setup or community adoption plan.